# Towards Balanced Strength and Plasticity in Graphene-Nickel Composites: The Role of Graphene, Bimodal Metal Powder and Processing Conditions

Olga Yu. Kurapova [1,2,*], Ivan V. Smirnov [2], Ivan Yu. Archakov [3], Chao Chen [4] and Vladimir G. Konakov [1,3]

[1] Laboratory of Mechanics of Novel Nanomaterials, Institute of Advanced Manufacturing Technologies, Peter the Great St. Petersburg Polytechnic University, Politecnicheskaya Str. 29, St. Petersburg 195251, Russia; vgkonakov@yandex.ru

[2] Saint Petersburg State University, Universitetskya emb. 7/9, St. Petersburg 199034, Russia; i.v.smirnov@spbu.ru

[3] Laboratory of Mechanics of Nanomaterials and Theory of Defects, Institute for Problems of Mechanical Engineering, Russian Academy of Sciences, Bolshoy pr. 61, St. Petersburg 199178, Russia; ivan.archakov@gmail.com

[4] State Key Laboratory of Powder Metallurgy, Central South University, Changsha 410083, China; pkhqchenchao@csu.edu.cn

* Correspondence: o.y.kurapova@spbu.ru

**Abstract:** Due to their higher strength and lighter weight compared to conventional metals, graphene-nickel (Gr-Ni) composites have recently gained growing interest for use in the automotive and aerospace industries. Homogeneous Gr dispersion, the metal powder dispersity and processing conditions play a key role in obtaining the desired grain size distribution, an amount of high angle grain boundaries thus reaching the desired balance between strength and plasticity of the composite. Here, we report an approach to fabricating graphene-nickel composites with balanced strength and ductility through the microstructure optimization of the nickel matrix. A graphite platelets (GP) content of 0.1–1 wt.% was used for the optimization of the mechanical properties of the material. In situ, conversion GP-to-Gr was performed during the milling step. This paper discusses the effect of bimodal nano- and micro-sized Ni (nNi and mNi) on the mechanical properties and microstructure of Gr-Ni composites synthesized using a modified powder metallurgy approach. Specimens with varied nNi:mNi ratios were produced by two-step compaction and investigated by X-ray diffraction (XRD), scanning electron microscopy (SEM), transmission electron microscopy (TEM), Raman spectroscopy, electron back-scattered diffraction (EBSD) and nanoindentation. The best combination of ultimate tensile strength (UTS), yield limit (YL), elongation and hardness were obtained for 100nNi and 50nNi matrices, and the best composites were those with 0.1% graphene. The addition of more than 0.5 wt.% GP to the nickel matrix induces the fracture mechanism change from tensile to brittle fracture. Dedicated to the 300th anniversary of the St. Petersburg University Foundation.

**Keywords:** nickel-matrix composites; graphene; bimodal precursor powder; ultimate tensile strength; yield limit; elongation; Vickers hardness; modified powder metallurgy





## 1. Introduction

The interest in the new generation of metal matrix composites (MMCs) development has been growing staidly over the past decade. Due to their enhanced toughness, strength and thermal and electrical properties, MMCs often demonstrate a better performance than conventional metals and their alloys [1,2]. Among them, graphene-reinforced nickel-based (Gr-Ni) composites should be pointed out because of their superior mechanical, electrical and thermal properties. The excellent mechanical characteristics of graphene (tensile strength ~130 GPa and Young modulus ~1 TPa, see, e.g., [3], outstanding electro- and thermal conductivity [4–7]) coupled with their light weight, high strength, toughness

and good corrosion resistance make Gr-Ni composites the most promising materials for various engineering and biomedical applications [8,9].

The reinforcement by graphene has been proven to have an advantage over the other commonly used nano-inclusions such as carbon nanotubes (CNTs) [10–14], oxide nanoparticles [15,16], carbides [17,18] and nitrides [19,20]. As follows from [21], the major problems of graphene-doped MMCs are as follows: (i) the agglomeration of graphene during composite synthesis; (ii) weak interfacial bonding between Gr and the metal matrix and (iii) poor structural integrity of graphene. Graphene is coherent with nickel, enabling stronger bonding between graphene and the nickel metal matrix in Ni-Gr composites compared to the other Gr-reinforced MMCs [22–25]. It enables stress transfer between graphene and the Ni matrix and contributes to overall reinforcement efficiency.

In turn, the homogeneous Gr distribution in the metal matrix also has a great impact on the load transfer efficiency and the possibility to reach enhanced strength while maintaining good ductility [21]. It can enhance or, in the case of graphene agglomeration, weaken the strength of the nickel-based composite. In addition, the type of the Gr-containing additive (graphene nanoplatelets; GNPs, graphene oxide; GO, reduced graphene oxide; rGO, or Gr itself) and the production approach [26,27] are among the key factors affecting Gr distribution. In the recent work of authors [28], the effect of the graphene derivative type on the mechanical properties of nickel-graphene composites has been investigated. It was shown that a minimum amount of 0.1 wt.% Gr in the form of rGO and GNPs is enough to provide very different mechanical properties of composites.

So far, several studies have been conducted to fabricate graphene-reinforced nickel composites with superior properties [25,27–34]. As mentioned in [35], the high solubility of carbon in Ni and the formation of carbides require the implementation of complex sintering approaches such as friction stir processing [36], electrochemical deposition [37–40], spark plasma sintering (SPS) [8,24], selective laser sintering (SLS) [18,41], etc. In the work of C. Zhao [8], a complex approach consisting of molecular-level mixing followed by SPS was used to produce Ni-Gr specimens with a homogeneous graphene distribution. It was shown that the addition of 1.5 wt.% of Gr-containing additive resulted in a tensile strength increase of up to 95.2% and a yield strength increase of up to 327.6%. along with a sufficient elongation (12.1%). In situ, high-temperature Chemical Vapor Deposition (CVD) followed by SPS applied by K. Fu et al. in [27] resulted in the formation of a 3D graphene network that hinders grain growth and significantly improves the composite microstructure. The yield strength of 474 MPa and tensile strength of 546 MPa were obtained for the composite containing 1.0 vol.% of Gr additive. The obtained composites, although exhibiting superior hardness, are limited by the geometry of obtained samples and are not likely to be mass-produced.

Recently, several feasible and scalable approaches for nickel-graphene composite fabrication based on modified powder metallurgy were suggested in [30,35,42,43]. For instance, Zhang et al. [35] fabricated a Ni-Ni$_3$C composite with a nacre-like, brick-and-mortar structure using Ni powders and graphene sheets. The composite achieved a 73% increase in strength with a 28% compromise on ductility, leading to a notable improvement in toughness. The 0.1 wt.% GNPs-Ni composites obtained in [42] via the powder metallurgy and spark plasma sintering (SPS) showed the improvement of yield strength by 29.5% and ultimate tensile strength by 24.8%, respectively, and preserved good ductility. J. Jiang with co-authors [44] synthesized bulk Ni-Gr composites by graphene in situ growth in the nickel matrix during the powder metallurgy procedure; the authors reported the composite hardness and the tensile strength of ~107 HV and 370 MPa, 1.7 and 4.1 times that of pure Ni, respectively.

As seen, the described approaches were shown to be efficient in achieving improved strength and good ductility in Ni-Gr composites through the reinforcements by 0.1–1.5 wt.% Gr for most composites. Obviously, a key point here is that the ratio of the particle surfaces of the MMC and Gr additive provides a measure of grain size control without undesirable graphene agglomeration, see, e.g., [45]. Indeed, the type of Gr derivative used for reinforce-

ment determines the effective surface of the Gr additive, while the synthesis procedure provides the uniformity of the Gr-additive distribution and preservation of Gr-integrity. At the same time, the effective surface can be also controlled through the variable particle size of nickel powder (nano-, micron-sized powders). The use of nickel powders with different dispersities at certain processing conditions may provide a bimodality of grain sizes in the nickel matrix and significantly contribute to the enhancement of mechanical properties. As it was demonstrated in [46], the introduction of graphene into the MMC under several manufacturing conditions also can lead to the bimodality of the grain size distribution of the metal matrix. The obtained nanocomposites reinforced by 1.6 vol.% graphene nanoplatelets exhibited Young's modulus, yield strength and ultimate tensile strength of 55 GPa, 271 MPa and 352 MPa composite. The values were improved by 20%, 166% and 35%, respectively, compared with that of the unreinforced magnesium-based alloy due to the control of grain growth by graphene. Thus, the impacts of graphene, metal particle size and the processing conditions and their synergetic effect on the mechanical properties of composites are not clear and should be investigated in detail. The present research continues the series of works [28,30] devoted to the Ni-Gr composites fabrication with superior properties. Therefore, the aim of the present study was the investigation of the effect of a bimodal matrix, graphene addition and processing conditions on the microstructures, mechanical properties and fracture mechanisms of the Gr-Ni composites. The novelty of the approach consists of the use of two different nickel matrices, nano and micro-sized, which are mixed in different percentages, as well as a second compaction step that gives the composite outstandingly different mechanical properties.

## 2. Materials and Methods

### 2.1. Materials and Synthesis Procedure

Three series of samples were taken for investigation (Table 1). The samples of series I were produced using the mixtures of nano-and micron-sized nickel powders (nNi and mNi, respectively) with the nNi:mNi ratios of 100:0, 85:15, 65:35, 50:50, 35:65, 15:85 and 0:100 without any carbon derivative addition. The samples of Series II were produced from the powder mixtures of 100nNi + x wt.% of graphene addition (x = 0; 0.1; 0.2; 0.5 and 1.0) without mNi. Series III utilizes the mixtures of 50 wt.% nNi–50 wt.% mNi with x wt.% of graphene (x = 0; 0.1; 0.2; 0.5 and 1.0).

**Table 1.** Specimens and their abbreviations.

| Series | Composition of the Powder Mixture, wt.% | Abbreviation for Final Ni-Gr Composite |
|--------|------------------------------------------|-----------------------------------------|
| | 100mNi–0nNi | 100mNi |
| | 85mNi–15nNi | 15nNi |
| | 65mNi–35nNi | 35nNi |
| I | 50mNi–50nNi | 50nNi |
| | 35mNi–65nNi | 65nNi |
| | 15mNi–85nNi | 85nNi |
| | 0mNi–100nNi | 100nNi |
| | 0.1Gr–99.9nNi | 0.1Gr–100nNi |
| II | 0.2Gr–99.8nNi | 0.2Gr–100nNi |
| | 0.5Gr–99.5nNi | 0.5Gr–100nNi |
| | 1.0Gr–99.0nNi | 1Gr–100nNi |
| | 0.1Gr–99.9(50mNi–50nNi) | 0.1Gr–50nNi |
| III | 0.2Gr–99.8(50mNi–50nNi) | 0.2Gr–50nNi |
| | 0.5Gr–99.5(50mNi–50nNi) | 0.5Gr–50nNi |
| | 1.0Gr–99.0(50mNi–50nNi) | 1Gr–50nNi |

Nanosized nickel powder (mean particle size 80 nm, purity ≥ 99.76%, "Advanced Powder Technologies", Ltd., Tomsk, Russia) and micron-sized nickel powder (commercially available PNE-1 powder with the mean particle size of 20 μm and Ni contents ≥ 99.50%) were used as the starting materials for sample fabrication. The microstructures of the initial

Ni powders are shown in Figure S1. Commercial graphite, intercalated by ammonia ("Activ nano", Ltd., Saint Petersburg, Russia) was used as a graphene precursor. In order to obtain graphite platelets (GP), commercial graphite was thermally exfoliated at 600 °C for 10 min for further in situ conversion into graphene. The microstructure of the produced GP is presented in Figure 1.

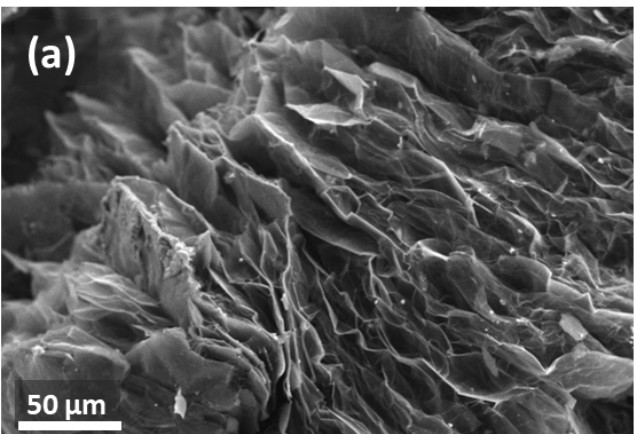 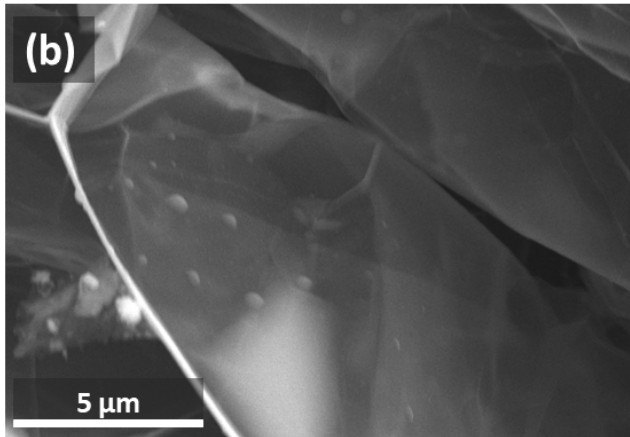

**Figure 1.** High-resolution SEM photos of graphite platelets (GP) after thermal exfoliation at 600 °C; (**a**) magnification ×500; (**b**) magnification ×10,000.

As seen from the Figures, microstructures of thermally exfoliated GP represent a stack of layers that are connected with each other. Each of them is wrinkled and veil-like (see, Figure 1b). All the compositions were mixed by dry ball-milling under optimized conditions (Pulverisete-6 planetary mill (Fritsch, Hamburg, Germany), 400 rpm for 2 h with 2 min reverse cycles), which created an in situ graphite-to-graphene conversion [30,47]. Milling was performed in a $N_2$ atmosphere with a water and oxygen content less than 0.001 vol.% to prevent nNi oxidation because of its high flammability in air. The obtained composite powders were compacted into green body pellets with a diameter of 25 mm and a height of 9 mm using cold uniaxial pressing (12.5 t/cm$^2$, 15 min). Obtained green bodies of series I specimens (see Table 1) were then subjected to hydrostatic pressing at 152 MPa. All the obtained samples were annealed in a vacuum at 1250 °C for 1 h. Thus, three series of metallic composites were obtained, and their abbreviations are presented in Table 1.

*2.2. Characterization*

X-ray diffraction analysis (XRD, SHIMADZU XRD-6000 (SHIMADZU, Kyoto, Japan), Cu-Kα at λ = 1.54 Å) was used to identify the phase composition of the specimens. Raman spectroscopy (Raman Spectrometer Horiba SENTERRA T64000 (Horiba Corp., Kyoto, Japan); the wavelength of the excitation laser was 488 nm) was used to investigate the state of carbon allotropes.

Scanning electron microscopy (SEM, Supra 55VP, Carl Zeiss QEC GmbH, Peine, Germany), energy-dispersive X-ray spectroscopy analysis (EDX, Oxford Instruments INCAx-act X-ray microanalysis spectrometer, Oxford Instruments, Wycombe, Buckinghamshire, UK) and electron backscatter diffraction technique (EBSD analysis, TESCAN MIRA 3LMH FEG with EBSD unit "Channel 5", Hitachi S-3400 N, Hitachi, Ltd., Kyoto, Japan) were used to study the nickel specimens and composites microstructures. The EBSD procedure was carried out with the accuracy of misorientation angle and axis determination less than 2 and 5°, respectively. Patterns of backscattered electrons (EBSP) were acquired from a rectangular grid with a step size of 0.5 μm for 23 μsec per one EBSP acquisition during mapping. Oxford Instruments Aztec HKL analysis software version 6.0 (Oxford Instruments, Wycombe, Buckinghamshire, UK) was used to identify the crystal orientation from the Kikuchi pattern in automatic mode, and the crystal structure of metallic nickel was used to estimate the grain size in the composites. The minimal determined grain size was

3 pixels; grain boundaries were determined as the linear intercepts between high-angle grain boundaries. SEM and EBSD data were obtained from the surfaces of the samples. The samples for SEM and EBSD were first embedded into an epoxy resin. Then, the surfaces were polished with a series of microcrystalline diamond suspensions and finished with a nano-sized colloidal silica suspension. No etching was performed. For TEM, an extremely thin cut (<100 μm) was taken from a sample embedded in hardened resin. The cut was double-polished to ensure the absence of scratches from the cut. Then, the polishing material was removed by repeated washing in water and alcohol. This allows for the electron beam to pass from the electron gun through the specimen to the detector. The apparent densities of specimens were measured by hydrostatic weighting (scales RADWAG 220 c/xc, Radwag, Radom, Poland). Each sample was weighed in air and then in isopropyl alcohol. The data were obtained for 3 samples and averaged over 5 independent measurements for each sample. Vickers hardness tests (Shimadzu HMV-G21DT, SHIMADZU, Kyoto, Japan) were performed using a diamond pyramidal indenter with a 2 N load (HV0.2) applied for 15 s (data was averaged over 15 tests over the specimen's cross-section). Mechanical properties tests were also carried out via uniaxial tension tests using a SHIMADZU AG X-Plus test machine (SHIMADZU, Kyoto, Japan) at the strain rate of $10^{-3}$ s$^{-1}$. Each curve was averaged over 5 tests. The t-bone-shaped samples (shoulder blade) were cut along the cross-section of the specimens using an electrical erosion machine. The size of the working part of the samples was 6 mm in length, 2 mm in width and 1.3 mm in thickness. Fractography (Zeiss Auriga Laser, Carl Zeiss QEC GmbH, Peine, Germany) was used to perform the failure analysis of specimens.

## 3. Results and Discussion

### 3.1. Microstructure and Mechanical Properties of Nickel Specimens Manufactured from mNi and nNi Powders

To provide a better understanding of the ratio of graphene to metal matrix effects on the final properties of composites, microstructure and mechanical properties of the reference specimens—specimens produced from the powder mixtures with 0, 15, 35, 50, 65, 85 and 100 wt.% of nano-Ni—were investigated. Typical inverted polar figures (IPF) images obtained by EBSD for the above specimens are shown in Figure 2. As seen from the figure, the number of dislocations is negligible in all specimens, while the grain boundaries are clearly defined. All specimens can be characterized by a high amount of high-angle grain boundaries (HABs) and twins. Specimens with the maximum nano-Ni contents (100nNi and 85nNi) also demonstrated significant grain growth during the manufacturing process. Note that the mean size of the initial nano-powder particles was ~80 nm; therefore, a three times magnitude growth can be observed.

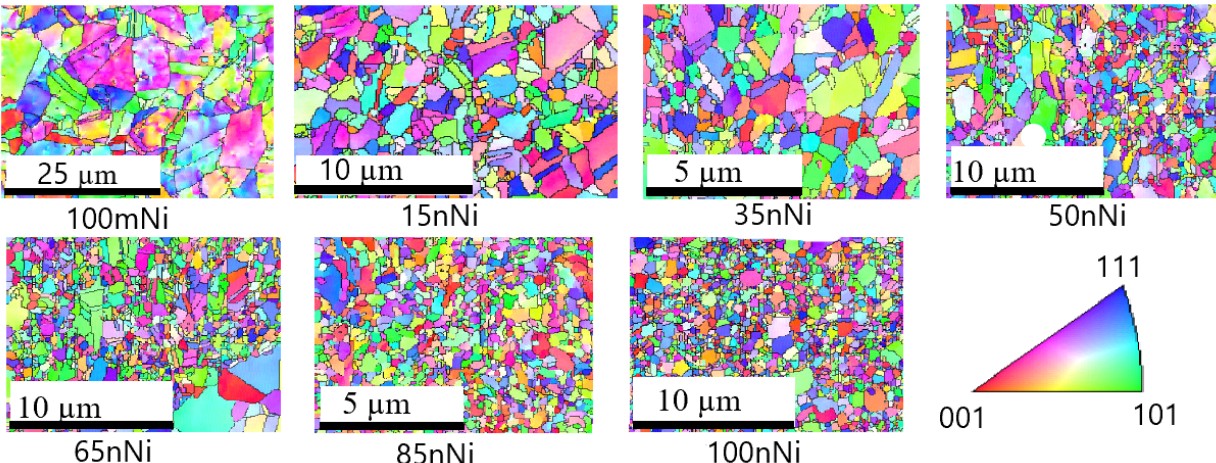

**Figure 2.** Microstructures of the specimens of Series I; the comparison of the IPF images obtained for Series I via EBSD; HABs are marked by black lines.

The information on the grain size and grain boundaries type was obtained from the analysis of the images taken from the large surfaces (more than 5000 grains for each specimen); the grain boundaries were subdivided into the following groups: misorientations below 15° were treated as LABs (low-angle grain boundaries), special misorientations that could be interpreted as being due to the coinciding site lattices were attributed to CSLs (coincident site lattices) and all others were treated as HABs (high-angle grain boundaries). Since CSL misorientation also exceeds 15°, data includes the CSL fraction. Figure 3 demonstrates the dependencies of the mean grain size (here, grain size was estimated as the diameter of a circle having the same area as the grain area from EBSD data) and the HABs fraction. Grain size distributions are shown in Figure S2, Supplementary Materials.

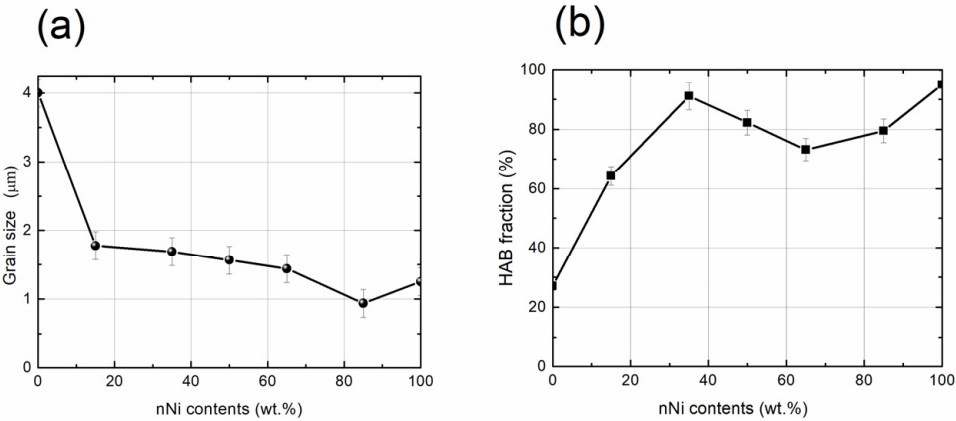

**Figure 3.** (**a**) Mean grain size and (**b**) HABs fraction as a function of nano-Ni contents.

As seen from Figure 3a, the grain size in the specimens decreases with the increase in nNi contents in the initial precursor; however, some slight increase in the mean size is observed for 100nNi specimen. As for the HABs fraction, it increases with nNi contents up to 35% (HABs fraction here exceeds 90%); a further increase in nNi contents resulted in some decrease in the HABs fraction (down to ~75% at 65% of nNi) followed by the increase of up to ~95% for the pure nNi specimen. Table 2 summarizes the data on grain boundary types in the specimens studied. It should be noted that the specimen with 35% nNi is characterized by the maximum fraction of CSL, while pure nNi showed the maximum HABs fraction.

**Table 2.** Analysis of the grain boundary types in the specimens studied.

| Specimen | Grain Boundary Fraction, % | | |
|---|---|---|---|
| | LAB | CSL | HAB |
| 100mNi | 72.9 | 5.9 | 21.2 |
| 15nNi | 35.7 | 25.7 | 38.6 |
| 35nNi | 8.9 | 40.4 | 50.7 |
| 50nNi | 17.8 | 30.7 | 51.5 |
| 65nNi | 26.8 | 29.8 | 43.4 |
| 85nNi | 10.5 | 10.7 | 68.8 |
| 100nNi | 5.0 | 28.6 | 66.40 |

The data on the specimen's density and Vickers hardness are presented in Figure 4. As seen from the figure, the densities of the specimens manufactured using a powder metallurgy approach are somehow lower than that determined for the density of bulk Ni (8.9 g/cm$^3$). The specimens' density changes close-to-linear within the experimental error, exhibiting a local minimum for the 35nNi specimen. This might be due to the presence of

numerous defects in the microstructure caused by the irregular packing of the particles during the initial compaction of the metal powder. Despite only the 85nNi specimen consisting of submicron grains (mean grain size is $0.65 \pm 0.05$ μm), it is characterized by a rather low amount of HAB and CLS fractions (31.2 and 10.7%, see Table 2). The values are close to the ones obtained for 100mNi specimens. The latter shows the lowest HAB and CLS fractions as well as the coarser grains and lower hardness values. The 50nNi specimen is characterized by its balanced properties. In contrast to the other specimens of series I, the grain size distribution in the 50nNi specimen is close to bimodal (see Figure S2). Analyzing the above results, the use of Ni powder having a complex fraction composition (50nNi–50mNi) is prospective for further reinforcement using graphene.

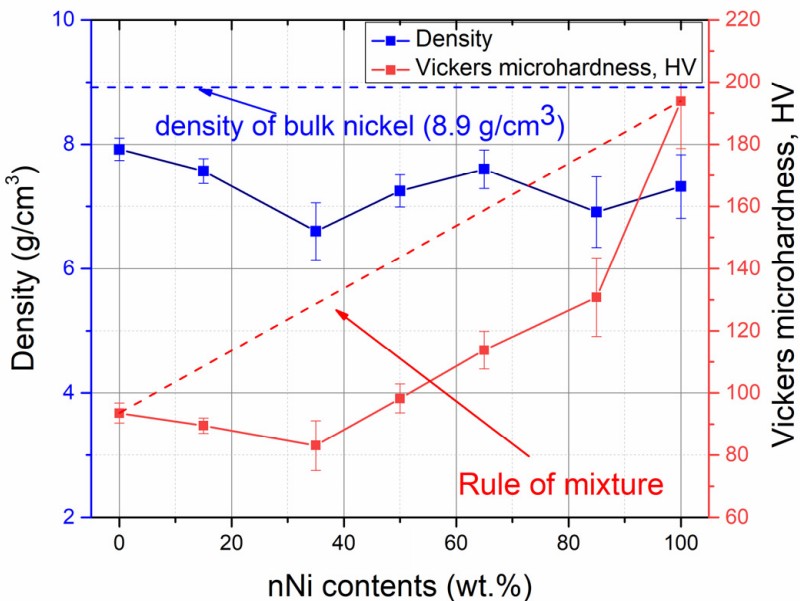

**Figure 4.** The dependencies of specimen densities and Vickers hardness on nano-Ni contents.

### 3.2. The Structure of Nickel-Graphene Composites

The X-ray patterns obtained for series II and III specimens are shown in Figures 5 and 6.

As seen, the phase composition of the 100nNi specimen corresponds to the face-centered cubic structure (FCC) typical for metallic nickel. The peak at $2\Theta = 26.5°$ corresponding to carbon is absent in the XRD patterns obtained for the composites containing 0.1–1.0 wt.% Gr. Remarkably, the intensity ratio of the peaks at $2\Theta = 44$ and $51°$ in the XRD pattern of 0.5Gr–nNi is slightly changed. For the specimens in series III, which were manufactured using a 50nNi powder, the phase composition corresponds to metallic nickel with no other admixtures (see Figure 6). As seen from Figures 5 and 6, a considerable variation of intensity ratio is seen just for one specimen (0.5Gr–100nNi). Since the intensity ratios are not changing considerably (no enhancement of the peak at $2\Theta = 76°$), it can be concluded that no texturing of the sample takes place upon the incorporation of GP. Along with the absence of a peak at $2\Theta = 26.5°$, this gives the conclusion that a homogeneous distribution of graphene in the metal matrix was reached. The change of the intensities is likely due to the geometry of the sample exposed to XRD (the surface roughness contributes to the resulted diffraction pattern).

The states of the carbon allotropes in the Gr-Ni composites fabricated using different powder dispersities were investigated via Raman spectroscopy, see Figure 7. Spectra obtained for the starting powder of thermally exfoliated GP, 100nNi and 50mNi specimens are presented for a comparison. Additionally, TEM analysis was carried out for 0.1Gr–nNi and 1Gr–nNi composites (see Figure 8).

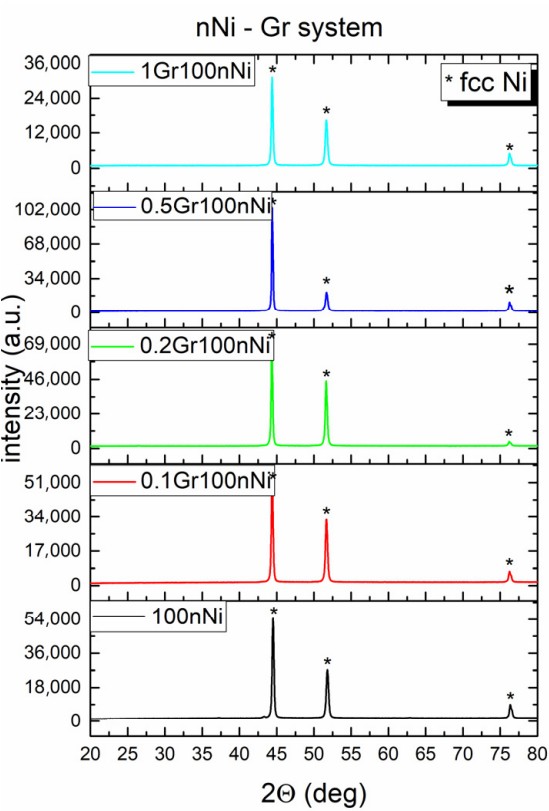

**Figure 5.** The X-ray patterns obtained for the 100nNi specimen and the 0.1Gr–nNi, 0.2Gr–nNi, 0.5Gr–nNi and 1Gr–nNi composites (specimen series II).

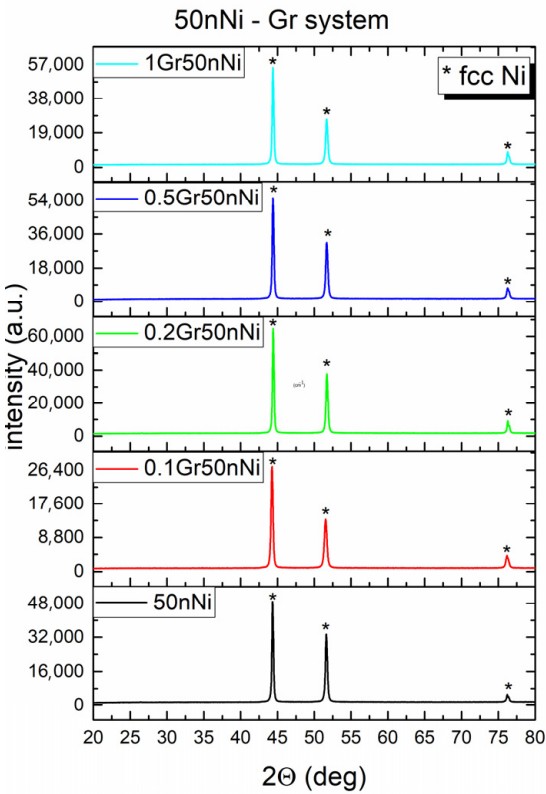

**Figure 6.** The X-ray patterns obtained for the 50nNi specimen and 0.1Gr–50nNi, 0.2Gr–50nNi, 0.5Gr–50nNi and 1Gr–50nNi composites (specimens of the series III).

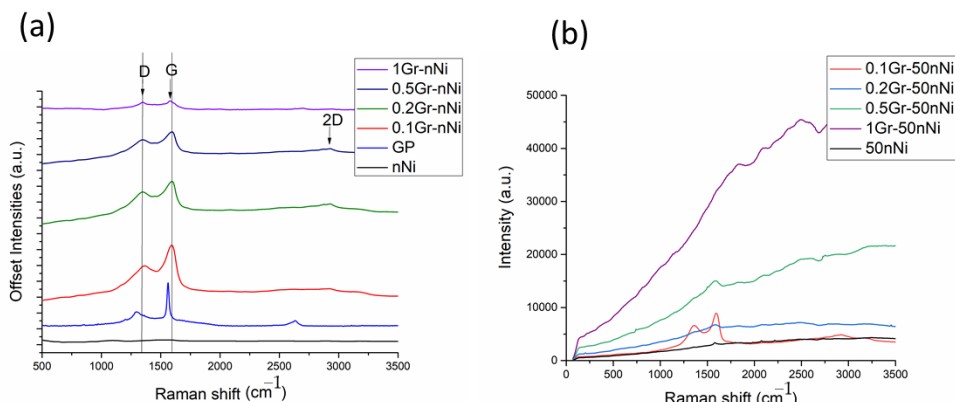

**Figure 7.** Raman spectra obtained for (**a**) series II composites and (**b**) series III composites in comparison with the data for thermally exfoliated GP, 100nNi and 50nNi.

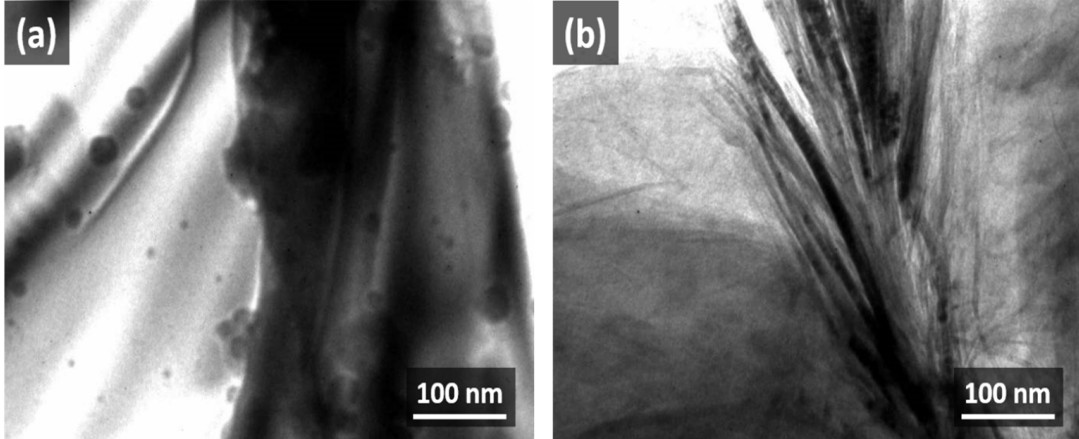

**Figure 8.** (**a**) The light-field TEM photo of a graphene flake in 0.1G–nNi composite; (**b**) the light-field TEM photo of a graphene flake in 1Gr–nNi composite.

As seen from the spectra obtained, the spectrum obtained for thermally exfoliated GP contained D, G and 2D bands at 1297, 1565 and 2635 cm$^{-1}$. Their positions and intensities differ from the ones in the spectrum of graphite [47]. Thermal exfoliation efficiently results in the formation of a microstructure characterized by veil-like layers stuck together in one GP block (see, Figure 1), resulting in the Raman shift of the bands compared to graphite. In particular, the intensity of the D band in the spectrum obtained for GP, corresponding to the breathing modes of sp$^2$ atoms in rings, is higher than the one for graphite. The characteristic bands of Gr (D, G and 2D at ~1378, 1585 and 2880 cm$^{-1}$, respectively [47]) are present in the spectra of Gr–100nNi composites. Thus, in situ GP-to-Gr conversion takes place during the manufacturing process for both series II and III. The D and G peaks are slightly overlapped, being a typical picture for nickel matrices [8,9,30]. In the case of the 1Gr–nNi composite, the positions of the D and G bands are shifted. Here, the G band is a superposition of two peaks at 1580 and 1626 cm$^{-1}$, indicating the presence of a certain amount of graphite in the composite. The $I_D/I_G$ ratios obtained for 0.1Gr–nNi, 0.2Gr–nNi and 0.5Gr–nNi composites are less than one, being ~0.67, ~0.80 and ~0.82 respectively (see Table 3). The addition of 1 wt.% GP results in an almost equal $I_D/I_G$ ratio of 0.99 in the spectrum obtained for the 1Gr–nNi composite. For the composites of series III, a rather strong luminescence of the nickel matrix is observed (see Figure 6b). Both D and G bands are present only in the spectrum of the 0.1Gr–50nNi composite with the ratio being close to 0.1Gr–nNi one. Only the G band is seen at 1585 cm$^{-1}$ in the spectra of 0.2Gr–50nNi and 0.5Gr–50nNi composites. The 1Gr–50nNi shows the spectra of luminescence. All $I_D/I_G$ ratios obtained are less than one, so one can conclude that no structural damage took place

during the manufacturing of the composites. However, the addition of 1 wt.% GP results in the graphite admixture in the composite. The results of the Raman spectroscopy analysis are in good accordance with the TEM data presented in Figure 8.

**Table 3.** The mechanical properties of graphene-nickel specimens and $I_D/I_G$ ratio were assessed from Raman spectra.

| Composite | Ultimate Tensile Strength, UTS (MPa) | Yield Limit (MPa) | Uniform Elongation (%) | Maximum Elongation (%) | HV (-) | $I_D/I_G$ Ratio |
|---|---|---|---|---|---|---|
| 100nNi | 203 ± 1 | 117 ± 7 | 4.1 ± 0.6 | 5.8 ± 0.4 | 99 ± 8 | - |
| 0.1Gr–nNi | 187 ± 10 | 119 ± 5 | 3.5 ± 0.1 | 3.9 ± 0.2 | 100 ± 4 | 0.67 |
| 0.2Gr–nNi | 202 ± 16 | 114 ± 4 | 4.7 ± 0.9 | 5.4 ± 0.8 | 95 ± 10 | 0.80 |
| 0.5Gr–nNi | 53 ± 12 | 52 ± 12 | 0.2 ± 0.1 | 0.6 ± 0.3 | 81 ± 4 | 0.82 |
| 1Gr–nNi | 52 ± 3 | - | 0.2 ± 0.03 | 0.2 ± 0.03 | 52 ± 11 | 0.99 |
| 50nNi | 366 ± 9 | 186 ± 3 | 19 ± 1 | 20.4 ± 1.5 | 101 ± 12 | - |
| 0.1Gr–50nNi | 193 ± 33 | 91 ± 6 | 17 ± 2 | 19.8 ± 1.3 | 79 ± 8 | 0.67 |
| 0.2Gr–50nNi | 126 ± 7 | 98 ± 4 | 1.6 ± 0.3 | 2.7 ± 0.8 | 81 ± 6 | - |
| 0.5Gr–50nNi | 73 ± 5 | 71 ± 4 | 0.4 ± 0.1 | 0.6 ± 0.2 | 75 ± 15 | - |
| 1Gr–50nNi | 60 ± 3 | - | 0.1 ± 0.03 | 0.3 ± 0.06 | 97 ± 13 | - |

The light-field TEM demonstrates that graphene in the 0.1Gr–nNi composite is present in the form of wrinkled flakes and consists of one or two layers. The multi-layered flakes are seen in the TEM image for the 1Gr–nNi composite, confirming graphite presence (see, Figure 8b). The obtained Raman and TEM data shows that complete GP-to-graphene conversion during ball milling of 1 wt.% Gr-100nNi composite powder at 400 rpm is not reached. The conclusion is in accordance with our recent work [30]. Figures 9 and 10 compare the microstructures of metallic specimens fabricated from the 100nNi and 50nNi powders with the composites from Series II and III reinforced with 0.1 wt.% Gr.

Figures 9 and 10 compare the microstructures of the composites of Series I fabricated from the 100nNi and 50nNi with the composites from Series II and III with 0.1 wt.% Gr.

As seen, the addition of 0.1 wt.% graphene does not significantly change the microstructure of 50nNi and 100nNi specimens. Nevertheless, some minor modifications could be assumed: the pores close and become lightly smaller; the elongated pores transform to spherical ones in the case of the 0.1Gr–50nNi specimen. Note that the pores are homogeneously distributed on the 0.1Gr–100nNi and 0.1Gr–50nNi composites surfaces. EBSD data provides more detailed information on the microstructures of the specimens. The addition of 0.1 wt.% GP to nanosized nickel powder results in the hindering of grain growth, and the formation of individual large grains is suppressed. The 0.1Gr–100nNi composite possesses a more homogeneous structure composed of submicron-sized and micron-sized grains. In the case of the initially bimodal powder (50nNi), the incorporation of 0.1 wt.% GP amount also results in a certain grain refinement. The allocated areas of grains with smaller grains and larger sizes (all micron-sized) are well distinguished. The grain distributions in 0.1Gr–100nNi and 0.1Gr–50nNi are close to bimodal.

The typical microstructures of composites synthesized from the powders with different graphene addition are shown in Figures 11 and 12.

The introduction of 0.2 wt.% GP into the 100nNi matrix results in a slight porosity decrease in the composite, however, the grain sizes enlarge considerably. Further increase of GP content to 0.5 wt.% leads to a significant pores size increase and their transformation to round-shaped pores (0.5Gr–100nNi composite). The maximum contents of 1 wt.% GP leads to pores merging with the formation of pores agglomerates of irregular shape allocated along the grain boundaries. Large grains of ~50 μm are well distinguished for 1Gr–100nNi

composites (see Figure 12b). The incorporation of 1 wt.% GP to 50nNi matrix results in the formation of elongated pore agglomerates and fast grain coarsening (see Figures 11f and 12c). Grains with sizes close to ~500 µm, separated by smaller grains of ~50 µm, are present in the structure of the 1Gr–50nNi composite. Coupling SEM, TEM and EBSD data, one can expect much lower mechanical properties of the 0.5Gr–100nNi, 1Gr–100nNi and 1Gr–50nNi composites than those for composites with smaller graphene contents.

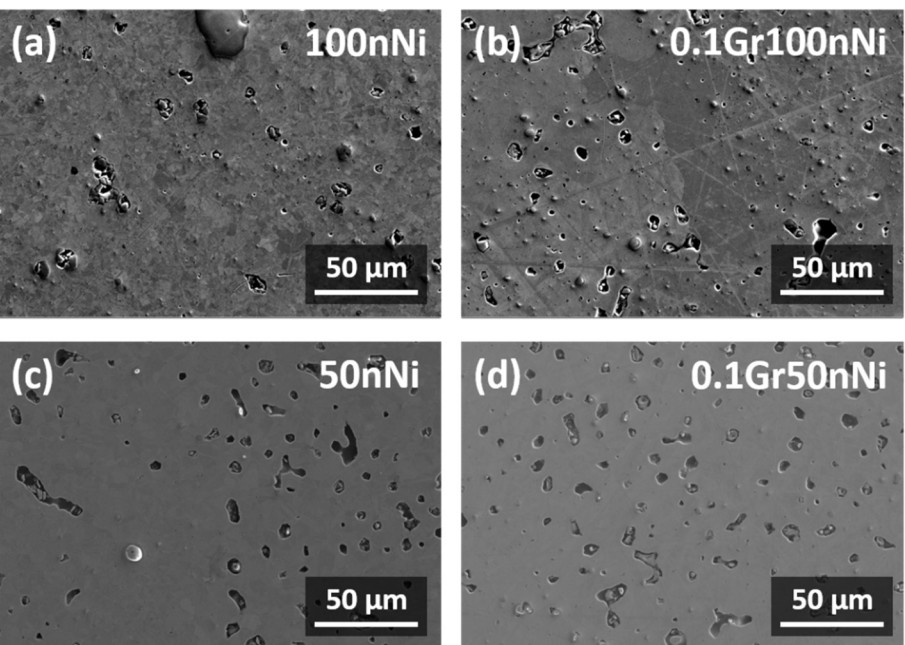

**Figure 9.** SEM data: the microstructure of (**a**) 100nNi, (**b**) 0.1Gr–100nNi, (**c**) 50nNi and (**d**) 0.1Gr–50nNi specimens. The striations are traces of polishing.

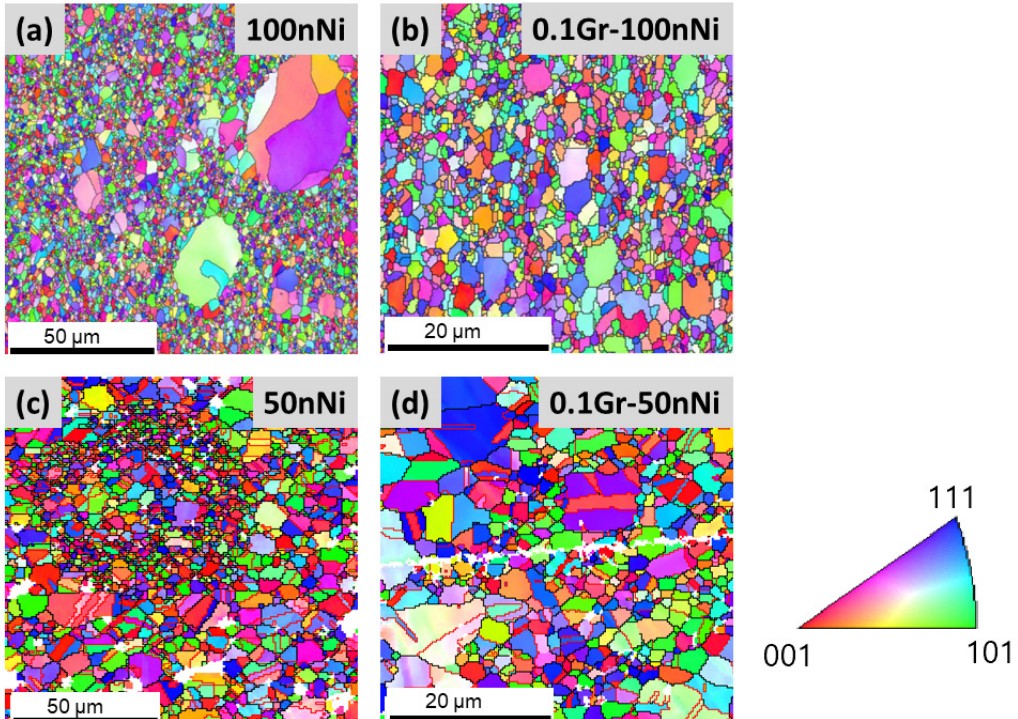

**Figure 10.** IPF maps obtained for (**a**) 100nNi specimen, (**b**) 0.1Gr–100nNi composite, (**c**) 50nNi specimen and (**d**) 0.1Gr–50nNi composite. The striations are traces of polishing.

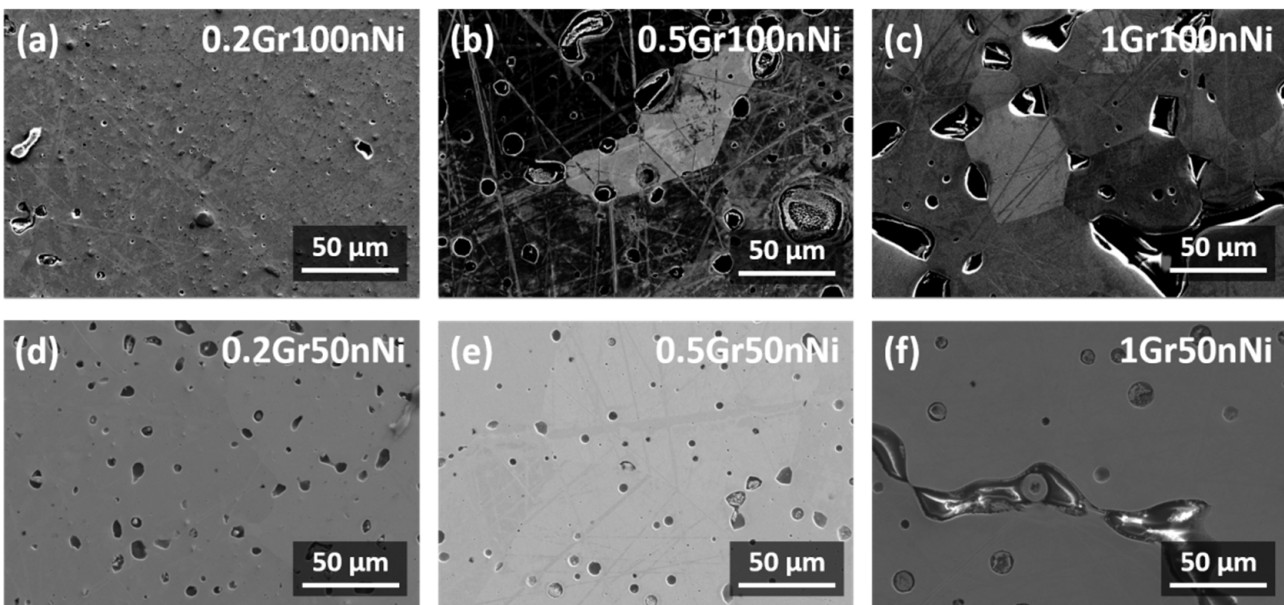

**Figure 11.** SEM data: the microstructure of (**a**) 0.2Gr–100nNi, (**b**) 0.5Gr–100nNi, (**c**) 1Gr–100nNi, (**d**) 0.2Gr–50nNi, (**e**) 0.5Gr–50nNi and (**f**) 1Gr–50nNi specimens.

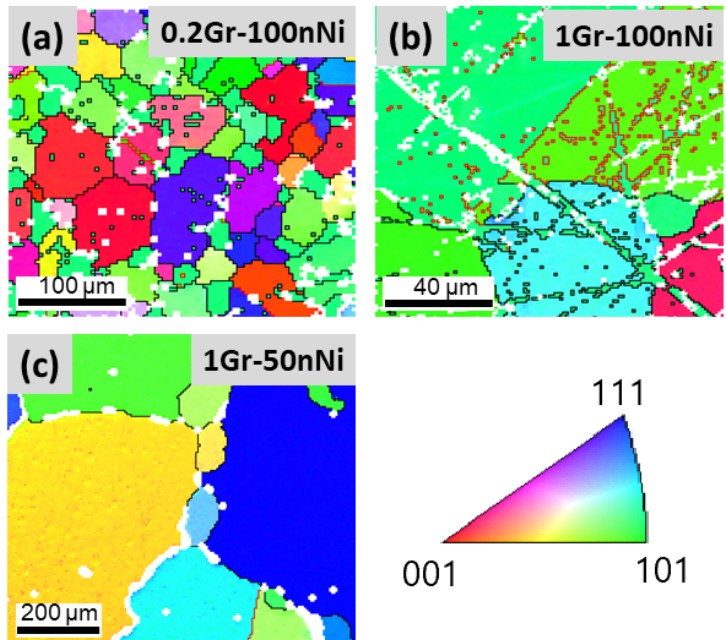

**Figure 12.** IPF images obtained for (**a**) 0.2Gr–100nNi and (**b**) 1Gr–100nNi composite under different resolutions; (**c**) 1Gr–50nNi composite.

### 3.3. The Mechanical Properties of Gr-Ni Composites

The stress-strain curves obtained for specimens of series II and III are shown in Figure 13.

As seen from Figure 13a, the mechanical properties of 100nNi, 0.1Gr–100nNi and 0.2Gr–100nNi exhibit close ultimate tensile strength (UTS) values and yield limit (YL). Among composites with a 100nNi matrix, the 0.2Gr–100nNi composite shows the best combination of uniform elongation and UTS and YL being close to 100nNi. The 0.1Gr–100nNi composite shows a slightly decreased ductility of its nickel matrix. The 0.1 wt.% GP incorporation results in about a factor of two decreases in UTS and yield limit values, while

the elongation remains almost the same, allowing for experimental error. The composites of both series II and III fabricated from powders having a graphene content of 0.5 and 1 wt.% demonstrate drastically decreased UTS and elongation values. The presence of pore agglomerates as well as the multi-layered graphene flakes in the grain boundaries result in the low bonding between the graphene additive and nickel matrix and the simultaneous decrease of strength and ductility.

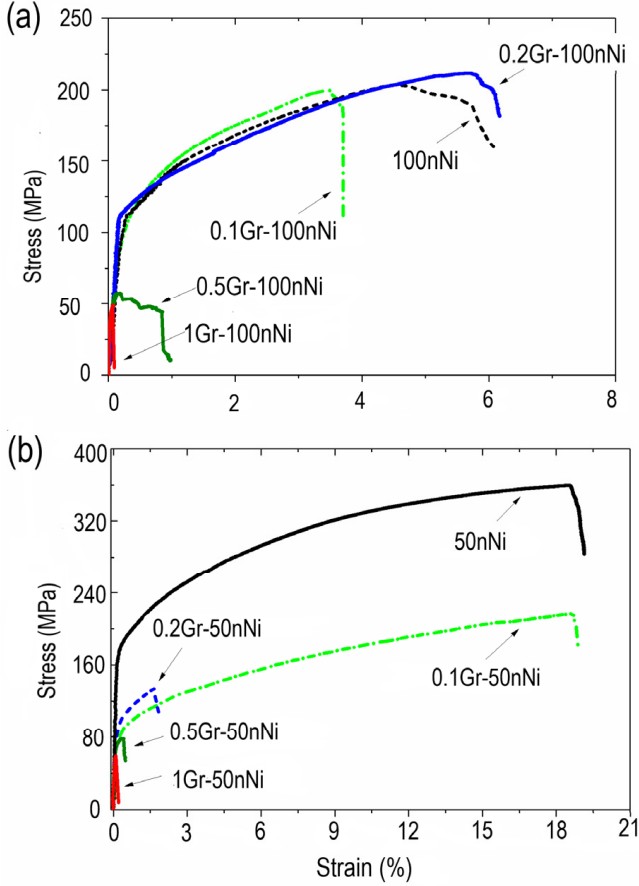

**Figure 13.** The stress-strain curves obtained for specimens of (**a**) series II and (**b**) series III.

Vickers hardness data is presented in Table 3 and Figure 14. The hardness values of reference samples 100nNi and 50 nNi, presented in Figures 4 and 14, differ by almost twice. It is due to the difference in the manufacturing approach: specimens of series I were manufactured using an additional step of hydrostatic pressing at 152 MPa, whereas series II and III were not made the same way. Comparing the Vickers hardness data of 100nNi specimen after two-step compaction and Gr–100nNi composites obtained, one can see that the values of hardness are also twice decreased when uniaxial compaction is used. Increased hardness of the specimens after additional processing are likely due to the same strengthening mechanism related to grain sizes. Refined microstructure (absence of pores and large microstructure defects) leads to an increase of hardness values. The hardness of Gr–Ni composites manufactured from nNi powder show the gradual decrease of values with the increase of GP content from 0.2 to 1 wt.%. In contrast, specimens of series III with 50nNi demonstrated a different behavior. As seen from Figure 14, the Vickers hardness of the reference specimen 50nNi manufactured with no additional compaction step is close to that of the composite with the maximum Gr contents (1Gr–50nNi), while the specimens 0.1Gr–50nNi, 0.2Gr–50nNi and 0.5Gr–50nNi are characterized by somehow lower hardness, which is nearly the same for these specimens.

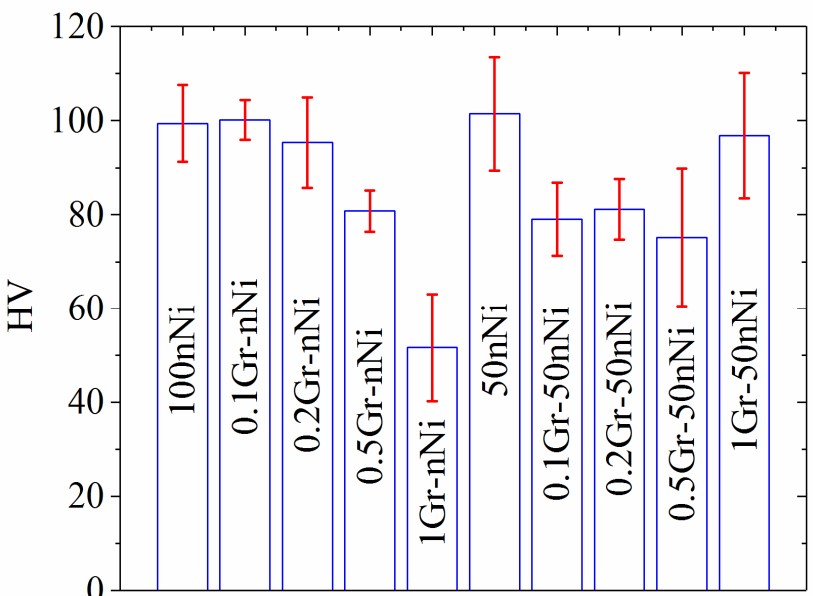

**Figure 14.** The comparison of Vickers hardness obtained for 100nNi and 50nNi specimens with the data for composites in series II and III.

### 3.4. Fracture of Nickel-Graphene Composites

The results of the mechanical testing are in accordance with obtained fractography data. Figures 15 and 16 represent the fractography test results, obtained for 100nNi, 50nNi specimens and composites produced from the powder mixtures with 0.1 and 1 wt.% Gr addition after the mechanical testing.

As seen from the figures, the fracture surface changes depending on graphene content. At the same time, the surface of all specimens contains particles of different sizes, resembling the shape of metallic nickel. Fracture surfaces, obtained for specimens with the same graphene content but different initial nickel precursor powder sizes, have similar features. However, they are different from those for 100nNi and 50nNi specimens (see, Figures 15 and 16a,b). The fracture surface of 100nNi composite consists primarily of shallow dimples of different sizes, where the particles are situated. The particles may act as the origin of the voids. Despite the more pronounced plastic deformation of the 50nNi specimen, dimple fracture is localized in disparate agglomerations along its fracture surface. The intergranular fracture process is expressed during the fracture of the 50nNi specimen, and it is followed by the secondary cracks. In addition, pronounced slip traces are observed on the grain boundaries. Thus, the plastic deformation of the two series of specimens is due to the different mechanisms. The specimens manufactured from bimodal nickel powder showed significantly higher tensile strength and elongation. The fracture surface relief becomes more uniform with the Gr amount increase. The dimple-like fracture contribution to the overall specimen fracture decreases when compared to the 100nNi sample. The brittle intergranular fracture starts to prevail. As seen in Figure 16e,f it is characterized by the pronounced block steps on the intergranular crack's surface. As discussed in [48], such an image of the fracture may be a result of the shear band's collision with a crack. The ledge formed favors the further brittle fracture process. One more mechanism can be due to the deformation in the region of the crack tip. Therefore, the traces of slips and twinning are left on the crack's surface. The observed step-like formations can take place on the grain surfaces or on the edge with pores or inclusions during the solidification and cooling of the material. This favors a surface energy decrease, when there is not enough material to fulfill the voids. The composites having 1 wt.% GP (1Gr–nNi and 1Gr–50nNi, see Figures 15 and 16e,f have smooth facets of the intergranular fracture with no typical fea-

tures of plastic deformation. The 1Gr–100nNi and 1Gr–50nNi specimens are characterized by the highest porosity value.

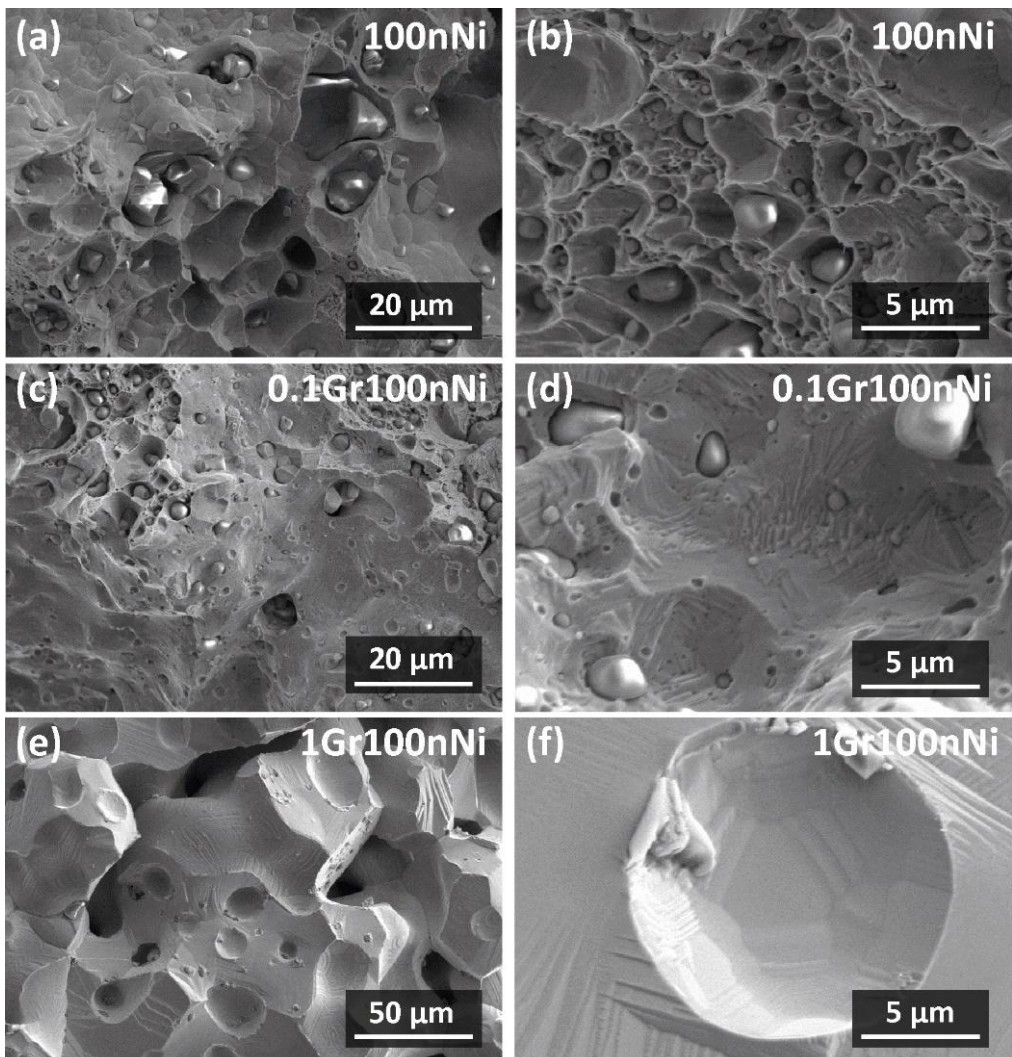

**Figure 15.** Fractography surfaces obtained for (**a**,**b**) 100nNi specimen and (**c**,**d**) 0.1Gr–100nNi composite, (**e**,**f**) 1Gr–100nNi after mechanical testing.

Therefore, the low tensile strength of the materials is due to the presence of the individual Ni particles in the structure and the low bonding of the multi-layered graphene flakes with the metallic matrix. The first microstructure feature, the presence of the particles, results in the decrease of the composite working section and increased local stresses. The second one, i.e., low bonding, does not provide a smooth transfer of stresses and deformations on the graphene-nickel interface.

The balance achieved in the mechanical properties of both Gr-Ni composites and specimens with the bimodal matrix with no Gr reinforcement, obviously, has different origins, or, in other words, different strengthening mechanisms. The two-step compaction contributes the most to the doubling of hardness values. The strengthening mechanism is likely due to the presence of high amounts of HAB and CLS fractions of the grain boundaries that act as pinning points to hinder dislocation propagation [35,49]. According to [50], the introduction of graphene results in the Young modulus enhancement for a pure metal (Ni or Al) only under compression. In the present work, tensile experiments were conducted. The obtained experimental results are in good accordance with a computational model proposed in [8] describing the mechanical properties of composites graphene-

ultrafine grained metal matrix and graphene-bimodal metal matrix. For composites with a homogeneous ultrafine-grained Ni matrix, grain-boundary sliding along graphene platelets can begin at the real flow stresses of the graphene-metal composites and reduce their yield strength. For composites having a bimodal structure of the metal matrix, a decrease in the yield strength of the composites is associated with grain boundary sliding. Indeed, when the graphene amount is increased to 0.5 and 1 wt.% steady embitterment of both 50nNi and 100nNi composites is seen. The mechanism proposed in [51] is confirmed by the fractography results (Figures 15 and 16). In this way, the optimal amounts of graphene addition to the Ni matrix for use in tensile conditions are 0.1–0.2 wt.%, which is in accordance with the available literature data on Ni-Gr composites manufactured via different techniques [8,26,27,42,52].

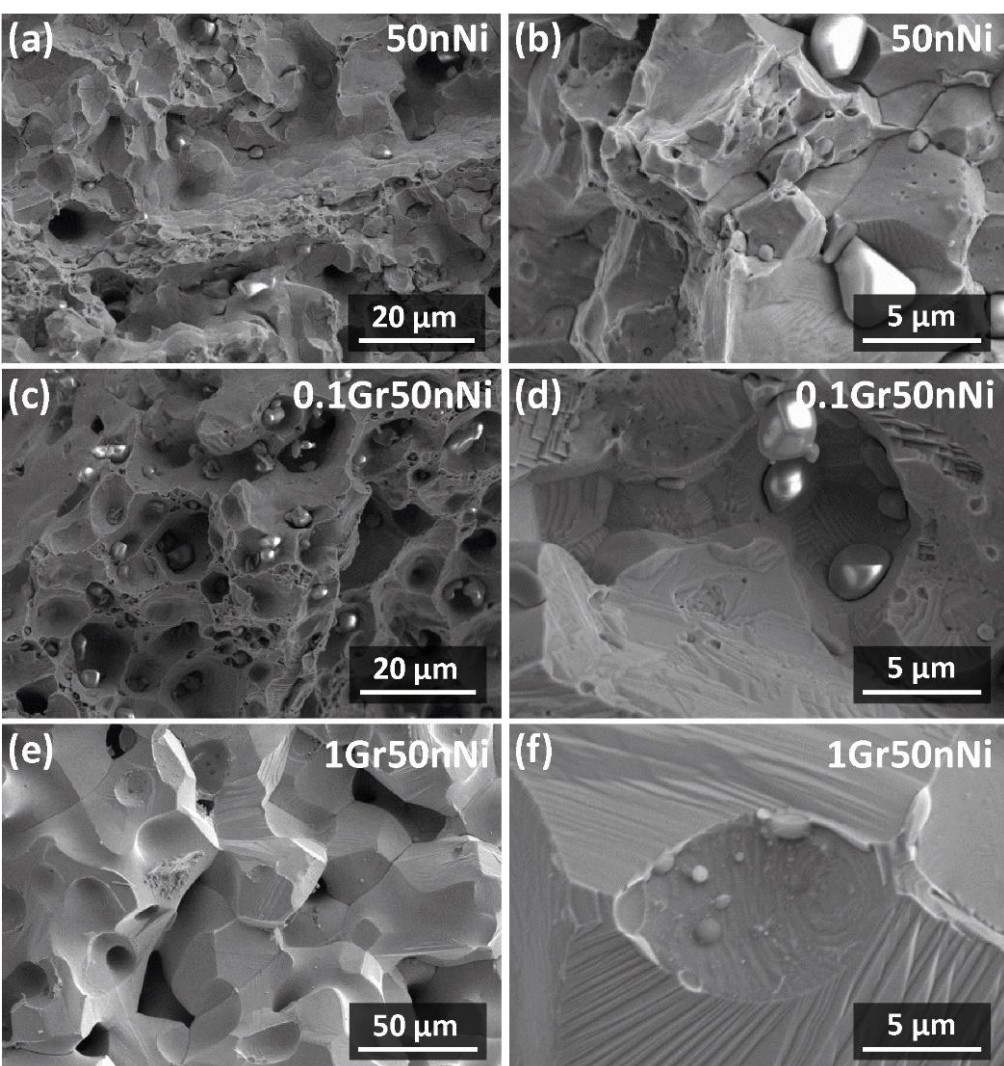

**Figure 16.** Fractography surfaces obtained for (**a**,**b**) 50nNi specimen and (**c**,**d**) 0.1Gr–50nNi composite, (**e**,**f**) 1Gr–50nNi after mechanical testing.

In summary, 100nNi, 50nNi and composites manufactured with 0.1 wt.% Gr using a modified powder metallurgy approach show a combination of balanced tensile strength and elongation. The use of the single step for powder compaction allows a rather facile Ni–Gr composite fabrication with sufficient mechanical properties. The introduction of an additional compaction step (hydrostatic pressing at 152 MPa) favors more uniform compaction leading to a doubling in hardness and can be used as an alternative to such an advanced technique, such as SPS.

## 4. Conclusions

Using mechanical testing, it was shown that 100nNi and 50nNi specimens exhibit the best combination of UTS, yield limit, elongation and hardness, and the best composites are those with 0.1wt.% graphene. The UTS values are $366 \pm 9$ MPa for the 50nNi specimen and $193 \pm 33$ MPa for the 0.1Gr-50nNi composite with the maximal elongation value of 20%. The step of hydrostatic pressing results in the doubling of hardness values for 50nNi and 100nNi materials. Via Raman spectroscopy and TEM it was shown that for 0.1–0.5 wt.% GP addition to Ni matrix, no structural damage took place during the manufacturing of the composites; the addition of 1 wt.% GP results in a graphite admixture in the composite. The addition of more than 0.5 wt.% Gr to nickel matrix induces the fracture mechanism change from tensile to brittle fracture.

**Supplementary Materials:** The following supporting information can be downloaded at: https://www.mdpi.com/article/10.3390/met13061037/s1, Figure S1: Microstructures of starting nickel powders: (a) and (b) micron-sized nickel powder; (c) and (d) nano-sized nickel powder; Figure S2: Grain size distribution in (a) 100mNi, (b) 15nNi, (c) 35nNi, (d) 50 nNi, (e) 65nNi, (f) 85nNi (g) 100nNi.

**Author Contributions:** Conceptualization, V.G.K. and C.C.; methodology, I.V.S. and V.G.K.; validation, O.Y.K. and I.V.S.; investigation, V.G.K., O.Y.K. and I.V.S.; resources, O.Y.K. and V.G.K.; writing—original draft preparation, O.Y.K. and I.V.S.; writing—review and editing, I.Y.A., I.V.S., V.G.K., C.C. and O.Y.K.; supervision, V.G.K.; funding acquisition, V.G.K. All authors have read and agreed to the published version of the manuscript.

**Funding:** The research was funded by the Ministry of Science and Higher Education of the Russian Federation as part of the World-Class Research Center program "Advanced Digital Technologies" (contract 075-15-2022-311).

**Data Availability Statement:** The data presented in this study are available on request from the corresponding author.

**Acknowledgments:** Raman spectroscopy data, EBSD and SEM fractography data tests were obtained at the research park of St. Petersburg State University «Center for Optical and Laser Materials Research», «Center for Geo-Environmental Research and Modeling (GEOMODEL)» and «Interdisciplinary Center for Nanotechnology», respectively.

**Conflicts of Interest:** The authors declare no conflict of interest.

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
