# Peer review of "Towards Balanced Strength and Plasticity in Graphene-Nickel Composites: The Role of Graphene, Bimodal Metal Powder and Processing Conditions"

_metals, doi:10.3390/met13061037_

Round 1
Reviewer 1 Report
This paper attempts to present the effect of using bimodal size Ni particles on Ni-Graphene (Ni-Gr) composites with different wt%.
Unfortunately, the data in the paper and in general the writing style as well as presentation need substantial improvement. The paper needs to be extensively re-drafted in terms of writing, data, figures, etc. to be understandable and coherent. Currently, there is a big disconnect between data shown and the discussion/conclusions.
Some key issues with the article are
1. The key findings of the paper are unclear and the various parts of the text contradict within themselves and with the figures/data. The claims are in general not backed up by the data.
a. At the core, data presented indicate that graphene addition is detrimental to mechanical performances. At best, the performances are maintained in some cases - not others by graphene addition. The claim on doubling of hardness (a key claim in the conclusions section) seems to be in comparison to a previously published work [Ref. 30?], with the actual previous data not presented in the current paper.
b. There is heavy inconsistency in the data.
For example, HV of 100nNi and 50nNi in Figure 13 and Table 3 is approximately the same ~99 and 101, respectively. But the data in Figure 3 is ~200 Hv for 100nNi and ~100 Hv for 50nNi.
2. The figures and the in-figure texts/labels as well as captions are inconsistent, confusing the readers. Examples -
a. The axes in Supplementary figure is hardly readable.
b. Scale bar values in figure 1 are not readable.
c. Figure 2 (a) and (b) figure captions dont match with the actual figure parts. The caption is swapped. There are no error bars - given that the data size is large - the data is from the analysis of 5000 grains!
d. Figure 3 - no error bars for the density
e. Line 225 claims "Two additional peaks of low intensity at 2Θ=39 and 43° are present in the pattern" - which cannot be seen in figure 4.
f. Raman spectra in Figure 6 - there is no Raman spectra of the original exfoliated graphite platelets (GP). But there are claims for graphite to graphene conversion.
g. The SEM and TEM of the original GP is not provided. From the TEM images, it is impossible to read the layer number in graphene (Figure 7).
h. The original SEM images of the Ni powders themselves are not provided.
i. It is not clear if the SEM images in Fig 8 and 9 are surface/cross section images, how these samples were prepared for SEM - any treatment to expose the surface, etc. is imissing.
j. Figure 8 and 9. Inconsistent labelling - the authors have assigned the label 0.1GrnNi for 100% nanoNi samples, but have labelled the samples here and elsewhere as 0.1Gr 100Ni.
k. Figure 10. In-figure labels for (b) and (d) are incorrect and dont match with the caption.
l. Figure 10 and 11. colour bar is missing.
3. The introduction does not sufficiently place the work in context of current literature. The experimental, results and discussion sections are insufficient in content, confusing and unclear. Examples:
a. The literature on the effect/role of metal grain sizes on Metal-Gr or Ni-Gr performances in context of processing and graphene (content) and addition is not clearly presented to justify the motivation of the current study.
For example: The authors state - "As it was demonstrated in [45], the introduction of graphene into the MMC under several manufacturing conditions also can lead to the bimodality of grain size distribution of the metal matrix. The obtained nanocomposites exhibited dramatically enhanced Young’s modulus, yield strength and ultimate tensile strength along with relatively high plasticity due to control of grain growth by graphene." But only vague terminology on "dramatically enhanced" properties is used without precise numbers or concrete discussion or additional references.
b. The experimental section is incomplete. Several particulars are missing - e.g.,
(i) Where were the GP purchased from? Or were they made in house?
(ii) How was the density measured, how many values were measured?
(iii) TEM and SEM sample preparations are missing.
c. Results/discussion:
The separation of results and discussion section makes it extremely hard to follow the data with 15 figures and 3 tables!
(i) No concrete explanation or discussion for
- deviation from rule of mixtures with regard to hardness values - figure 1
- variations in 2theta intensity ratios in XRD - Figure 4 and 5
(ii) What are the striations seen in SEM/IPF images - Fig. 8b, 9 (b, c, e), 10(d), 11(b)?
(iii) Data/data labels are incorrectly inferred in portions of text: Eg. Line 327: "As seen from Fig. 12(a) the mechanical properties of 100nNi, 0.1Gr-50nNi, 0.2Gr-50nNi composites are nearly the same." - is incorrect. As the data for 12a compares the 100nNi and 100nNi-Gr samples (series II) not 0.1Gr-50 nNi (series III).
The quality of English needs to be substantially improved. There are several typographical, grammatical and stylistic errors.
Reviewer 2 Report
The authors investigated the role of graphene, bimodal metal powder and 3 processing conditions on the microstructure and mechanical properties of Gr/Ni composite. I have some comments as below:
1. Microstructure of starting powder (mNi, nNi and Gr) should be provided.
2. High resolution SEM images of Gr/Ni powders and Gr/Ni composite should be presented, in which showing the presence of Gr.
3. Please explain why the formation of larger grain size and structural defects only observed with 35 nNi that leading to reducing the density of the composites compared to other samples?
4. Raman spectra of Gr should be provided to compare its characteristics with Gr/Ni powders.
5. The experimental results indicated that the best performance (hardness, tensile strength, elongation) was obtained with 50 nNi. The presence of Gr did not have any positive contributions on the performance of the composite? The presented results are quite opposite with other studies as using Gr as a reinforcement in the composite systems to improve their performance.
6. Strengthening mechanisms should be calculated to clarify the contribution of the components (mNi, nNi, Gr) on the mechanical properties of the composite.
7. The role of Gr on the balanced strength and plasticity in graphene-nickel composites seems not clear in the current manuscript.
Minor editing of English language required
Reviewer 3 Report
The manuscript describes a study of graphene-nickel composites, obtained by modified powder dispersion metallurgy, using small percentages of graphene. The novelty of the approach consists in the use of two different nickel matrices, nano and micro-sized, which mixed in different percentages, give the composites very different mechanical properties.
The article as a whole is well written and original, but there are some weaknesses that the authors need to address before it can be published.
Major issues
1. The use of acronyms without explicit terms is strongly discouraged in the abstract, but also in the text. Check that the first time you enter an abbreviation it is well described.
2. In lines 58-59, the sentence "It was shown that the mechanical behaviour of rGO-Ni and Gr-Ni composites differs at low additive contents." is very vague and it is not clear what the authors mean. If the concept is that a minimum amount of graphene material is enough to have very different mechanical properties, it should be rewritten.
3. In paragraph 2.1 the authors explain that they have used a further hydrostatic pressure of 152 MP on the series I of composites, those without graphene, but they do not explain the reason for this further treatment and how this procedure can modify the response of the materials. Please add a comment in the text, in the discussion.
4. Figure 1 is low resolution and out of focus. Please improve the image quality.
5. I find it unnecessary to use a supporting information file for just one figure showing grain size distributions, which are important in the discourse instead. Please the authors to insert it in the main text, and better describe the different distributions.
6. The caption of Figure 2 is not clear. Authors are asked to describe the graphs shown less cryptically, better in the text and not in the caption.
7. Table 2 lacks the error on the reported data. To have comparable information between the different samples it is necessary to enter the measurement error, otherwise those numbers have no meaning. The article loses enormously in precision and accuracy, please fill in the errors.
8. Table 2 also shows samples with total percentages of the fraction distribution much lower than 100% (100nNi sample) and slightly higher than 100% (85nNi sample). Authors should double-check the reported data and provide an explanation.
9. The references are incorrect, the work cannot be published. The references given on line 214 do not correspond to previous works by the same authors. Please check the bibliography carefully, this is a crucial point in all the articles.
10. In the text from lines 224 to 236 there are some omissions.
The low intensity peaks at 2Θ=39 and 43° are not evident in the pattern of 100nNi sample. The authors should include an enlarged figure of the 100nNi sample to highlight these nickel oxide peaks.
11. The description of Figure 4 is related to the II series, it would be better to make it explicit in the text, for the fluidity and legibility of the results.
12. The authors claim that the small graphene additive effectively suppresses the oxidation of nNi powder. But why doesn't the sample without 50nNi graphene show the presence of peaks related to the formation of the oxide (lines 235-236)? The authors need to clarify this point.
13. For a better understanding of the Raman spectra, a spectrum of the graphene-free 100nNi and 50nNi samples should also be included in Figure 6. Edit Figure 6. From what the authors said, only the 50nNi array should have luminescence problems, so I expect at least the 100nNi sample can be seen in Raman.
14. In the description of the samples at the SEM I find it a bit difficult to follow the authors' discourse, it would be appropriate to rewrite this part highlighting better the material on which the authors want to focus the reader's attention. They first showed images of the samples with 0.1% graphene (Figure 8), then the difference with a higher percentage of graphene. Then they focused on the 0.2% sample. Figure 10 then shows the IPF images of the 0.1% graphene samples. Greater data consistency is necessary and please rewrite this part in a more orderly manner.
15. The best combination of UTS, Yield limit, elongation and hardness were obtained for 100nNi and 50nNi matrices and the best composites are those with 0.1% graphene. These are the data that must emerge from the work and that must also be highlighted in the abstract. Please highlight these results in both the text and the abstract.
16. Conclusions are a boring list of sentences. Please rewrite this part following the suggestion just given.
Minor issues
In line 43 the term various referring to carbides is useless. It can be removed.
In line 62 the term “carides” needs to be corrected.
Reviewer 4 Report
Some references must be included:
1) Wang, X., Yu, M., Zhang, W., Zhang, B., & Dong, L. (2015). Synthesis and microwave absorption properties of graphene/nickel composite materials. Applied Physics A, 118, 1053-1058.
The novelty of this paper is not clearly emphasized.
The correlation between potential improved properties of this type of composite and method of synthesis is not clearly emphasized.
The conclusions must be reinforced.
Round 2
Reviewer 1 Report
The authors have made substantial improvement to the paper.
There are still some issues with the figure qualities and writing style, which need to be improved.
E.g.,
a. Please define nNi and mNi in the abstract before using these terminologies
b. Figure 6 - Y axis label is half cropped and half visible
c. Figure 10. Scale bar values are unclear
d. Font sizes and styles in the different figures are different - a uniform styling is recommended.
e. Figure S2 in the supplementary file is labelled also as S1.
Extensive English editing is still required. There are still many complex and awkward sentences. Although largely readable and understandable, the writing needs to be polished further.
Author Response
We thank the reviewer for the valuable time and careful reading of the manuscript.
a. nNi and mNi were defined in the abstract before using these terminologies, as suggested
b. Figure 6 - Y axis label is half cropped and half visible. Thank you. it is wierdly cropped indeed. We fixed it, thank you
c. Figure 10. Scale bar values are unclear
The scale bars were magnified to make them clear
d. Font sizes and styles in the different figures are different - a uniform styling is recommended.
The uniform Arial was chosen in all figures. The font size 26 was chosen for axes
e. Figure S2 in the supplementary file is labelled also as S1. Thank you. It was fixed.
Extensive English editing is still required. Thank you. The manuscript was spell checked and grammar checked, and English editing has been performed with the help of native speaker.
Reviewer 2 Report
The manuscript could be accepted in its current form.
Minor editing of English language required
Author Response
We thank the reviewer for a time and decision on the manuscript. The English was grammar and spell checked with the help of a native speaker.
Reviewer 4 Report
The paper was modified in accordance with the requests.
Author Response
We thank the reviewer for a time and valuable opignion.